# Volatiles from Different Instars of Honeybee Worker Larvae and Their Food

**DOI:** 10.3390/insects10040118

**Published:** 2019-04-25

**Authors:** Haohao Zhang, Chunsheng Hou, Pingli Dai, Yongjun Liu, Yanyan Wu, Yonggang Pang, Qingyun Diao

**Affiliations:** 1Key Laboratory of Pollinating Insect Biology, Institute of Apicultural Research, Chinese Academy of Agricultural Sciences, Beijing 100093, China; hhzhang83@163.com (H.Z.); houchunsheng@caas.cn (C.H.); daipingli@caas.cn (P.D.); liuyongjun@caas.cn (Y.L.); wuyanyan@caas.cn (Y.W.); 2Beijing Blooming Bio-tech Co., Ltd., Beijing 100079, China; pyg19831212@126.com

**Keywords:** *Apis mellifera*, GC-MS, larva, terpenoids, volatiles

## Abstract

(*E*)-β-Ocimene was the only volatile chemical found to be emitted by whole, live worker larvae of *Apis mellifera* L. when sampling in the vapor phase. In addition to (*E*)-β-ocimene, there is evidence for the existence of other volatiles, but the changes in their composition and contents remain unknown during larval development, as are their differences from larvae to larval food. We investigated volatile components of worker larvae and larval food using solid phase dynamic extraction (SPDE) coupled with gas chromatography-mass spectrometry (GC-MS). Nine compounds were identified with certainty and six tentatively, including terpenoids, aldehydes, hydrocarbons, an ester and a ketone. The contents of volatiles in the second-instar worker larvae differ greatly from those in larvae of other stages. This is mainly attributable to terpenoids, which resulted in the second-instar worker larvae having significantly higher amounts of overall volatiles. Larval food contained significantly higher amounts of aldehydes and hydrocarbons than the corresponding larvae from the fourth to fifth-instar. We discovered volatiles in worker larvae and their food that were never reported before; we also determined the content changes of these volatiles during larval development.

## 1. Introduction

Volatiles occur in honey bee colonies as a complex mixture, determined by pheromones produced by bees, and other chemicals emitted by beeswax, honey, pollen and larval food [1]. These volatiles have been identified as alcohols, aldehydes, benzenoid compounds, carboxylic acids, hydrocarbons, ketones and terpenoids [1].

Volatiles in honey bee colonies are easily received by bees through antennal reception, sometimes over long distances. Therefore, using volatiles is an efficient means to moderate bee social behaviors. With regard to the volatiles that maintain social cohesion, most come from adult worker bees, such as the alarm pheromone inducing defensive behavior of honeybees [2], the Nasanov pheromone triggering aggregation [3], and other volatiles that transmit recognition cues. A few volatiles are detected in queens specifically [4].

Very little effort has been made to determine the volatiles emitted directly by intact and alive honey bee larvae. In contrast, the nonvolatile chemicals of honey bee larvae are widely studied by solvent extraction methods [5,6,7]. (*E*)-β-Ocimene is the only identified volatile chemical produced by live worker larvae. There is evidence for the existence of other larval volatiles, because different chromatogram peaks in addition to (*E*)-β-ocimene were discovered [8], but their compositions are not clear. There is also a lack of information about how these unknown larval volatiles change with larval development. However, an understanding of the composition and content variation of volatiles during larval development is necessary for interpreting their functions.

Food provided to honeybees could directly or indirectly affect honeybee volatile production. There is evidence that food shortages might stimulate worker bee larvae to release more (*E*)-β-ocimene [9], and feeding honeybee worker larvae essential oils via diet supplements may change their volatiles [10]. However, the volatiles extracted from worker larval food have not been reported. Only volatile carboxylic acids were identified in drone larval food, and other unidentified non-acidic volatiles were noteworthy [11]. If the volatiles of the worker larval food were analyzed and combined with volatiles analysis of the worker larvae, it would provide a deep insight into the relationship between volatiles in larvae and their food during the same larval instar.

In this paper, we analyzed the volatiles from worker larvae and their food at different instars, using solid phase dynamic extraction (SPDE) combined with high resolution gas chromatography tandem mass spectrometry (GC-MS). Through this research, we discovered new volatiles associated with developing worker larvae and their food that have been previously overlooked.

## 2. Materials and Methods

Three standard Chinese commercial strains of Western honeybee (*Apis mellifera ligustica* L.) were used in this experiment. Large double-deep colonies (30,000 to 50,000 adult bees) were located at our apiary in the Institute of Apicultural Research in Beijing (39° N, 116.2° E). Queens were caged within plastic controllers (interior dimensions: 457 × 50 × 245 mm, only allowed for passage of worker bees) to oviposit on empty new combs (without pollen and nectar stores) for 24 h and were then removed to the outside of the controllers. Oviposition combs were kept in the controllers. At each caging interval, we obtained three combs with single-age cohorts of worker larvae. Caging events were carried out seven times, for a total of 21 combs.

After the combs were brought back to the laboratory, cells were randomly chosen to provide both the larval food and larvae samples for further analysis. Larvae and larval food was obtained with a small spatula. Larvae were inspected under the microscope, and only live and uninjured larvae were used for the study [9].

### 2.1. Treatments

The volatiles of grouped larvae at three development stages were monitored. Group sizes were 20 second-instar (2nd-instar) larvae, 10 fourth-instar (4th-instar) larvae, and 10 fifth-instar (5th-instar) larvae. The larval food volatiles were also detected at different stages. To monitor changes in volatiles with respect to ontogeny, the amount of volatiles released by individual larva was determined [8]. To compare volatiles content between larvae and food, the amount of volatiles released per unit weight of sample were calculated [12]. Larvae and food were carefully transferred into 20 mL glass vials. Before analysis, 2 µL of 0.1 mg/L hexadecane hexane solution on a strip of filter paper was added as an internal standard for quantification. Clean empty vials were analyzed separately as controls to remove background interference.

### 2.2. SPDE-GC-MS System

Volatile extraction was performed using SPDE (solid phase dynamic extraction) equipment installed in a CTC-Combi-PAL auto sampler (CTC Analytics, Zwingen, Switzerland), as described by Castro [13]. The SPDE needle (SPDE-01/AC-50-56, 50 µm × 56 mm), coated with 90% polydimethylsiloxane (PDMS) and 10% active charcoal (AC) was preconditioned before use. The equilibration between sample and headspace lasted for 30 min at 35 °C in an incubation pool. After equilibration, the extraction procedure occurred as follows: An extraction volume of 1 mL, agitator temperature of 35 °C, headspace syringe temperature of 35 °C, −30 strokes, and a filling/ejecting speed of 25 μL/s. During the equilibration and extraction procedure, the larvae were alive and isolated from food for more than 45 min [9]. The needle was then withdrawn and introduced into the injection port of the gas chromatograph, and pumped with 1 mL nitrogen at 100 μL/s for desorption and at 250 °C for 2 min in splitless mode.

GC-MS analysis was performed using a Shimadzu Gas Chromatograph-2010 equipped with a DB-5ms capillary column (30 m × 0.25 mm × 0.25 µm; J&W Scientific, Folsom, CA, USA) coupled to a Shimadzu Quadrupole-2010 Mass Spectrometer (Shimadzu, Kyoto, Japan). The oven program was as follows: 35 °C for 2 min, 35–200 °C at 5 °C/min, 200 °C for 2 min, 200–250 °C at 10 °C/min, then 250 °C for 3 min. Injector temperature was maintained at 250 °C, transfer line temperature was 250 °C, and ion source temperature was 200 °C. Helium was used as the carrier gas, at a flow rate of 1.7 mL/min.

### 2.3. Qualitative and Quantitative Analysis

The identification of the compound with authentic standards was performed by comparing the mass spectra (Wiley6 and NIST05) and retention times to those of authentic standards. Compounds without standards were identified by comparing the mass spectrum peaks with data system libraries (Wiley6 and NIST05) and other published spectra (Mass Spectrometry Data Centre 1974). Additionally, the linear retention indices (LRI) of the compounds were calculated by injecting a series of n-alkanes (C10–C25) (o2si Smart Solutions) into the GC-MS on two columns of different polarities under identical conditions. Authentic standards (listed in Appendix A) purchased from Alfa Aesar (Karlsruhe, Germany) were serially diluted with hexane to make standard solutions. The peak areas on the total ion chromatogram were used for quantification. The calibration curve derived from a step-series of standard compounds for individual target compounds was built by plotting the area ratio of target compounds to the internal standard against the concentration ratio. The concentrations of volatile compounds were calculated based on the corresponding calibration curves.

### 2.4. Statistical Analysis

Principal component analysis (PCA) using a correlation matrix was applied to the data to establish relationships between the different samples and their volatile compounds. The SPSS software package Version 21 for Windows (SPSS, Chicago, IL, USA) was used for statistical analysis. A one-way ANOVA (*p* < 0.05) was used to test for significant differences in volatile compound concentrations among treatments. Figures were created using SigmaPlot version 12 for Windows (Systat Software Inc., San Jose, CA, USA).

## 3. Results

We detected fifteen compounds from the developing larvae and their corresponding food, which could be sorted into seven groups: three aldehydes, one ester, three hydrocarbons, one ketone, and seven terpenoids (Figure 1, Table 1).

### 3.1. Volatiles

#### 3.1.1. Terpenoids

Terpenoids were the largest group collected from larvae except the 5th instar, accounting for 83.3%, and 73.7% of total larval volatiles at the 2nd and 4th instar, respectively, calculated as ng/mg. The major terpenoids of food volatiles were α-cedrene (t5) and cedrol (t7), accounting for 65.5–89.2% of the total terpenoids, while the major terpenoid of larvae was (*E*)-β-ocimene (t2), accounting for 60.9–93.8% of the total terpenoids.

The larval weight increased with larval development (Figure 2), but there was a continuous decrease in (*E*)-*β*-ocimene (Figure 3, column t2), α-terpineol (Figure 3, column t4), and total terpenoids (Figure 3, column St) when the amount of terpenoids was calculated as ng/individual larva. Two compounds disappeared completely at the 5th instar: (*Z*)-β-ocimene (Figure 3, column t1) and (*E*, *Z*)-alloocimene (Figure 3, column t3). No significant differences were observed among instars for α-cedrene (Figure 3, column t5) and β-cedrene (Figure 3, column t6). The amount of cedrol (Figure 3, column t7) increased sharply from the 2nd to 4th instar, and it remained stable from the 4th to 5th instar.

When the terpenoid amount (St, t1, t2, t3, t4, t5, t6, t7) released by the larva was calculated as ng/mg (Figure 3), there was a decreasing trend during larval development. The 2nd instar larvae always had significantly higher contents than larvae at other instars. When the terpenoid amount in larval food was calculated as ng/mg, the detectable terpenoid amount changed insignificantly among instars.

When comparing the content in larvae and food at the same stage as ng/mg, the larvae had a significantly higher amount of terpenoid than food at the 2nd instar, excepting α-cedrene, β-cedrene and cedrol. At the 2nd instar, the contents of α-cedrene and β-cedrene were insignificant between larva and food; the content of cedrol was significantly lower in larvae than in food. At the 4th instar, the larvae still had a significantly higher content of (*E*)-β-ocimene than food; the difference in (*Z*)-β-ocimene content was insignificant between larvae and food. For other terpenoid compounds, larvae had substantially lower contents than food after the 2nd instar.

#### 3.1.2. Aldehydes

Aldehydes were at higher concentrations in food (25.4–59.6%) than in larvae (7.9–22.3%) when calculated as ng/mg. Nonanal (a2) was the most abundant aldehyde in each sample, accounting for 65.3–93.3% of total aldehydes, followed by decanal (a3) and octanal (a1).

Octanal (Figure 3, column a1) was only detected in larvae at the 2nd instar, but it was present in food at every stage. Nonanal (Figure 3, column a2) and decanal (Figure 3, column a3) were detected in both larvae and food at every stage. When calculated by ng/individual larva, the content of each aldehyde and the total content of all aldehydes (Figure 3, column Sa) did not change significantly (*p* > 0.05) with the sharp increase in larval body weight (Figure 2). However, when calculated as ng/mg, both measures were significantly higher in larvae at the 2nd instar stage than at other stages. When comparing both measures in larvae and food at the same stage as ng/mg, there was no significant difference at the 2nd instar, while at other stages food contained higher aldehyde content than larvae (*p* < 0.01). The trends in aldehyde content in food were similar. There were fluctuations during larval development, but no significant differences occurred.

#### 3.1.3. Hydrocarbons

The amount of hydrocarbons compared to total volatiles content in food was the third highest after terpenoids and aldehydes, when calculated as ng/mg. Pentadecane (Figure 4, column h1) and heptadecane (Figure 4, column h2) were the hydrocarbons that accounted for the major number of hydrocarbons (69.7–88.1%).

When calculated by ng/individual larva, the hydrocarbon content generally increased during larval development. The 2nd instar larva had the lowest content of each hydrocarbon and the lowest total content of hydrocarbons in general. However, the situation was reversed when calculating as ng/mg. The 2nd instar larva had a significantly higher content than other instar larva, and there was an insignificant difference among other instars. Food at the 2nd instar also had a higher content than food at other instars, h2 significantly, h1 and h3 insignificantly. When comparing the content in larvae and food at the same stage by ng/mg, the differences were always significant, except at the 2nd instar. Larvae had a substantially lower content of hydrocarbon than food.

#### 3.1.4. Ester

Ethyl 2(*E*)-decenoate (Figure 4, column e1) was the only detected ester present before the 5th instar. This ester was significantly higher in 4th than 2nd instar larvae when calculated as ng/individual larva (*p* = 0.02). The trend was reversed when it was calculated as ng/mg. There were few changes in the food of different instars. At the 2nd instar, the larvae had a significantly higher content than food, but there was a significantly lower content at the 4th instar.

#### 3.1.5. Ketone

(*E*)-Geranylacetone (Figure 4, column k1) was the only ketone detected in this experiment. When calculated as ng/individual larva, the 5th instar larvae had significantly higher content than larvae at other instars, which had comparable levels. When calculated as ng/mg, the 2nd instar larvae had a significantly higher content than larvae at other instars. Food of various instars contained comparable levels of (*E*)-geranylacetone. When comparing the content in larvae and food at the same stage as ng/mg, larvae had substantially higher contents than food at the 2nd instar, while the opposite was true at other stages.

### 3.2. Principal Component Analysis

Principal component analysis (PCA) was performed to reveal relationships between the samples (scores) and their volatile compounds (loadings) (Figure 5). A total of 96 samples and 15 compounds were used (Kaiser-Meyer-Olkin Measure of Sampling Adequacy: 0.731; Bartlett’s Test of Sphericity: *p* < 0.01). The majority of information is contained in the first two PCs and accounts for 62.2% of the explained variation. A scatter plot of the PCA scores (Figure 5A) shows the distribution of samples, and the corresponding loading plot (Figure 5B) shows the volatiles in larvae and food at different development stages. Combining these results allows for interpretation of the relationships between the samples and their compounds.

The analyses show clear differences between the volatile profiles of larvae and the volatile profile of food. The two kinds of samples were separated by PC2. The majority of food samples had positive PC2 scores, attributable to the contribution of the aldehydes (a1, a2, a3), hydrocarbons (h1, h2, h3), and two of the terpenoids (t5, t7). Moreover, all of the larval samples had negative PC2 scores, driven by most of the terpenoids (t1, t2, t3, t4, t6), one ester (e1), one ketone (k1).

Samples of the same type were distinguished by PC1 according to development stages. Both larvae and food samples generally shifted from positive to negative along the PC1 axis, from the 2nd to 5th instar. Samples of 2nd-instar larvae located at the fourth quadrant were essentially characterized by the compounds distributed in that quadrant, especially terpenoids. Larvae at other instars were located in the third quadrant, implying that the volatiles of these stages were highly conserved and at relatively low levels. Part of the 2nd and 4th instars food located at the first quadrant was mainly characterized by the compounds distributed in that quadrant, especially aldehydes and hydrocarbons. The rest of the food samples were located in the third quadrant, with low PC scores.

## 4. Discussion

Terpenoids have previously been reported as characteristic products of the Nasanov gland of worker bees and include nerol, geraniol, (*E*) and (*Z*)-citral, nerolic acid, geranic acid and (*E*, *E*)-farnesol [14]. (*E*)-β-Ocimene can be emitted by established mated queens [4]. In the present study, we found that terpenoids are the major constituents of worker bee larvae. Larvae and food at the 2nd instar had a greater similarity of dry substance weight per unit volume compared to other stages. Assuming that larvae and food at the 2nd instar had similar densities, larvae would have released significantly higher amount of terpenoids than food; thus, these terpenoids, such as (*Z*)-β-ocimene, (*E*)-β-ocimene, (*E*, *Z*)-alloocimene and α-terpineol, were likely produced by the larvae. Some terpene synthases have been identified in larvae [9]. The terpenoid concentration could be diluted in old larvae because body fluid accounts for most of their weight. Under these conditions, terpenoid contents in larvae were still significantly greater than in food, suggesting that compounds such as (*Z*)-β-ocimene, (*E*)-β-ocimene (t2) and (*E*, *Z*)-alloocimene could be identified as larval volatiles. (*E*)-β-ocimene was determined to be a worker larval pheromone in a previous study [12]. (*Z*)-β-Ocimene and α-terpineol have not been reported as honeybee volatiles, but they have been indicated as pheromones in other insects. (*Z*)-β-Ocimene could be released by calling males of the Caribbean fruit fly (*Anastrepha suspensa*) [15]; α-Terpineol is reported as a compound of the male-produced aggregation pheromone secreted by the spined soldier bug (SSB) *Podisus maculiventris* [16].

(*Z*)-β-ocimene, (*E*)-β-ocimene and (*E*, *Z*)-alloocimene were once reported as the components of *Tagetes minuta* essential oil, which is highly lethal to *V. destructor* [17]. α-Terpineol was demonstrated to have a repellent effect on the mite in a laboratory assay, and may therefore repress its entry into the brood cells of hives [18]. These terpenoids might be noteworthy in *V. destructor* control.

There are sesquiterpenes identified as pheromones. (*E*)-b-Farnenese is an aphid alarm pheromone [19]. In this study, cedrol is a tentatively identified sesquiterpene with tertiary alcohol functionality. This alcohol could easily eliminate to give mixtures of α- and β-cedrene, these sesquiterpenes might be gathered by bees from gum and pollen of *Cedrus deodara* planted nearby the apiary.

All aldehydes detected in this study have been previously reported as the pheromone from calling males of *Galleria mellonella* [20] and the volatiles of hives. The aldehydes are emitted by adult worker bees [21] and virgin queens [22] and are present in enclosed brood combs containing active larvae and attending workers [1]. Our results found that aldehyde content per unit weight was higher in larval food than in larvae. This supports the assumption that aldehydes are secreted by worker bees, because larval food is a material that is manipulated by worker bees. This is consistent with the findings of Torto et al. [23] who reported these aldehydes in pollen. The result also explains why aldehyde contents in food did not show significant fluctuations during larval development. Aldehyde contents in larvae also remained steady, regardless of larval growth, except that a1 disappeared after the 2nd instar.

Hydrocarbons have transpiration-reducing functions in arthropods. They include linear and branched, saturated and unsaturated hydrocarbons, with different numbers of carbon atoms ranging from 15 to 35 [24]. Some of these compounds function in nestmate recognition and social acceptance [25]. Pentadecane has been reported as a larval volatile [26]. It was detected in nurse bees [27] and queens at liftoff [28], coupled with heptadecane. Octadecane is found in comb wax, with worker bees treated with this compound becoming less acceptable to their untreated nestmates [29]. It is released by the clearwing moth (*Paranthrene diaphana*) as female sex pheromone [30]. If we assume that cuticular hydrocarbons have an even and constant distribution on the surface of larvae, hydrocarbon content should increase with larval volume. A previous report has indicated that the level of tricosane and pentacosane would increase with drone larval development [31]. In the present study, hydrocarbon content increased at different degrees in individual larvae during larval development. To some extent, this provides another support for the hypothesis. In food, the hydrocarbon content showed different degrees of decrease in per unit weight. This suggests that the production of hydrocarbons in food is independent from that in larvae. In other words, hydrocarbons in food are affected by the food provider, while hydrocarbons in larvae are affected by the larvae themselves.

Aliphatic esters are known as another group of pheromones secreted by insects. For example, ethyl acetate is a male pheromone produced by the Mediterranean fruit fly (*Ceratitis capitate)* [32]. Acetate and propionate esters are found in the poison gland reservoir of *Myrmecina graminicola* [33]. In honeybee colonies, aliphatic ester is also ubiquitous. Decyl decanoate is secreted by virgin queens from the tergal gland; ethyl oleate is produced by forager bees, suppressing the onset of foraging among younger bees. Ethyl and methyl esters of palmitic, linoleic, linolenic, stearic, and oleic acids are brood pheromones mediating the communication between brood and worker bees [14]. Ethyl 2(*E*)-decenoate is tentatively identified in honeybees for the first time in the present study. Ethyl 2(*E*)-decenoate was once tentatively identified as one of the major compounds from fermented sugar baits for lepidopteran species resulting from sugar decomposition during the fermentation process [34].

(*E*)-Geranylacetone has been previously identified as a queen volatile [28], a male-produced sex pheromone of cerambycid beetle (*Hedypathes betulinus*) [35] and a male-produced aggregation pheromone of the brown spruce longhorn beetle (*Tetropium fuscum*) [36]. Concentrations of this compound in larvae did not show obvious changes with the growth of individual larvae, and the concentrations only displayed a dramatic increase during the capping stage (the 5th instar). No significant differences in the concentrations of this compound were found in food at different stages, either. This may suggest that the compound is released by larvae for a specific role in this particular period.

When tracing the origins of the volatiles of worker larvae and larval food, the following factors need to be considered in the future: the volatiles of the adult worker bees, the in-hive matrix (such as honey, pollen, propolis and wax), and even the out-hive plants published by other studies based on the same locality. Moreover, the constancy of volatile components in worker larvae samples from different localities should be established. Additionally, more qualitative analysis should be conducted to the tentatively identified volatiles. Finally, only a single extraction method was applied in this study, and the use of more advanced extraction or detection methods for worker larvae volatiles might yield additional compounds.

## 5. Conclusions

Our results show that volatiles could be identified from honeybee worker larvae and their food, in addition to (*E*)-β-ocimene. We provide evidence that these volatiles change and follow certain change rule during larval development. The present study should provide some basis for further research into the molecular mechanism of the volatiles, and for verification of the role of the identified components.

## Figures and Tables

**Figure 1 insects-10-00118-f001:**
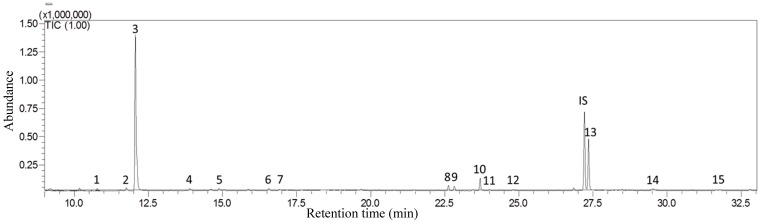
Gas chromatogram of all identified volatiles reflected in the 2nd instar larvae (IS: internal standard).

**Figure 2 insects-10-00118-f002:**
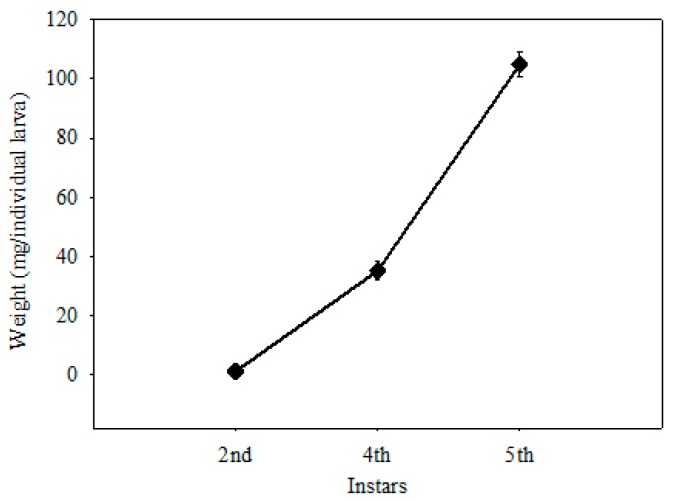
The changes of larva weight during larval development.

**Figure 3 insects-10-00118-f003:**
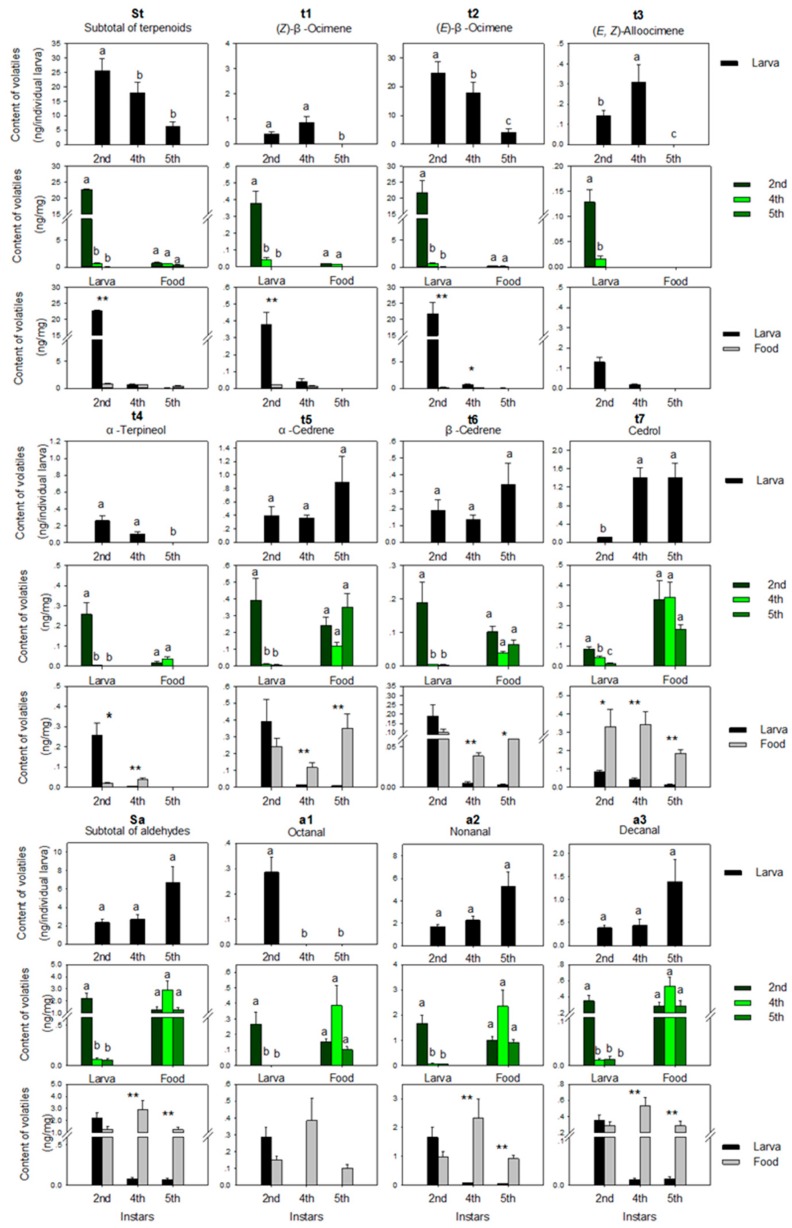
Terpenoids and aldehydes extracted from larvae and larval food at different larval instars. N > 10 for each group; error bars represent standard errors, different letters and ‘*’ on top of bars denote significant difference at the level of 0.05, ‘**’ at the level of 0.01 under Fisher’s PLSD test, after ANOVA showed a significant effect. Bars sharing a superscript letter are not significantly different.

**Figure 4 insects-10-00118-f004:**
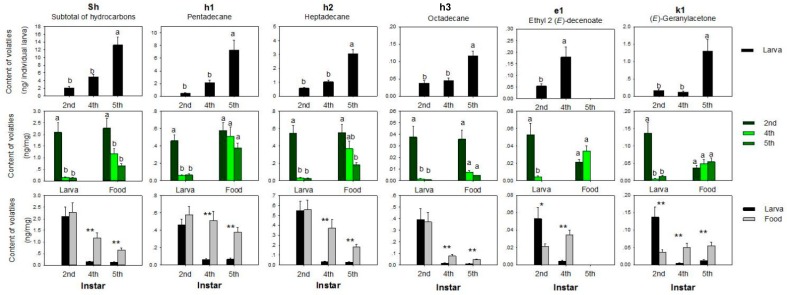
Hydrocarbons and other minor volatiles extracted from larvae and larval food at different larval instars. N > 10 for each group; error bars represent standard errors, different letters and ‘*’ on top of bars denote significant difference at the level of 0.05, ‘**’ at the level of 0.01 under Fisher’s PLSD test, after ANOVA showed a significant effect. Bars sharing a superscript letter are not significantly different.

**Figure 5 insects-10-00118-f005:**
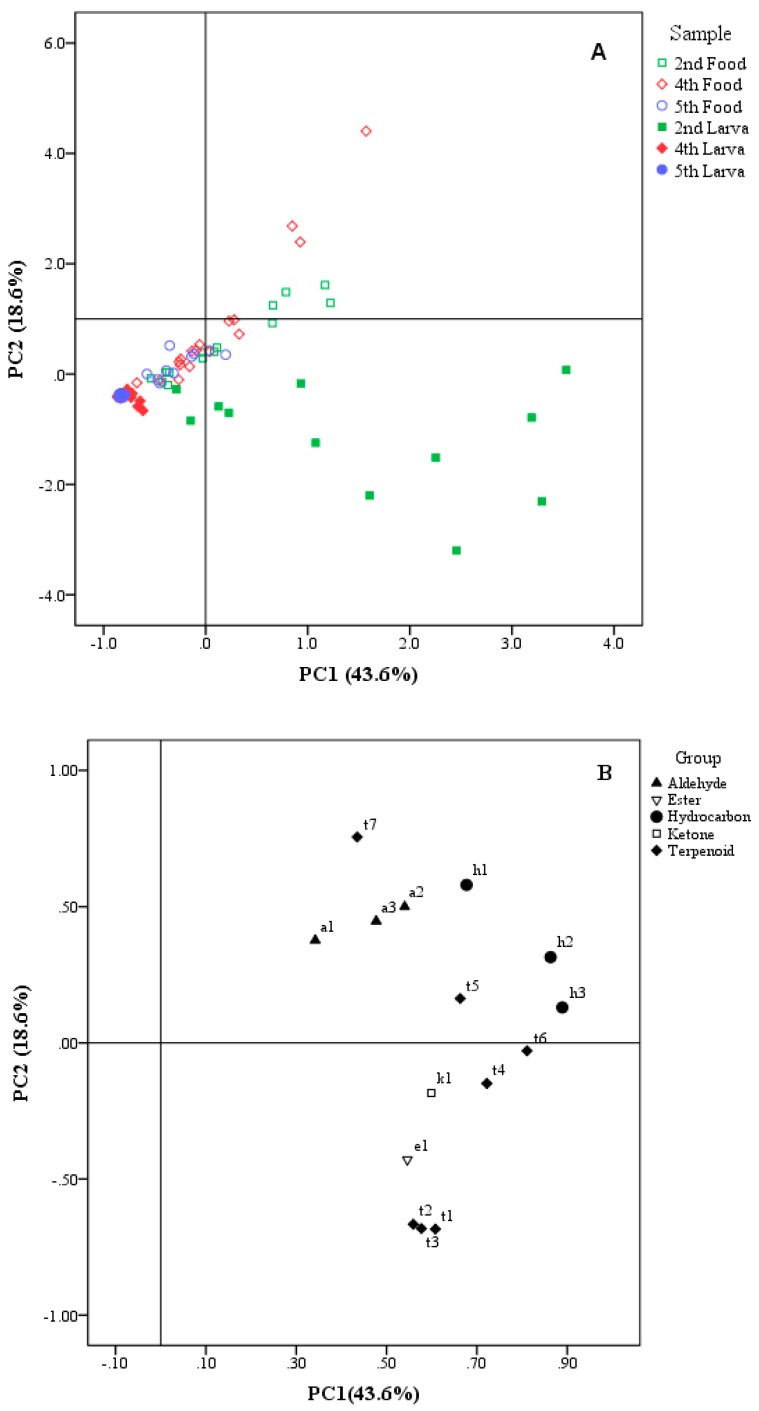
Positions of PC scores of samples including larvae and larval food at different larval instars (2nd: 2nd instar, 4th: 4th instar, and 5th: 5th instar). (**A**): Scores scatter plot of principal component analysis (PCA). (**B**): Loadings plot of PCA. The codes in (**B**) correspond to the compound codes listed in Table 1.

**Table 1 insects-10-00118-t001:** Volatile compounds tentatively identified from larvae and larval food at different larval instars.

RT ^a^	Compound	Code ^b^	Peak	LRI Calc ^c^		LRI Lit ^d^	Identify ^e^	Diagnostic Ions
				DB5	BPX5			
	Terpenoids							
11.75	(*Z*)-β-Ocimene	t1	2	1040	1229	976	S. N. L	93, 41, 79
12.06	(*E*)-β-Ocimene	t2	3	1050	1250	976	S. N. L	93, 79, 105
14.61	(*E, Z*)-Alloocimene	t3	5	1134	1371	1088	N. L	119, 91, 134
16.56	α-Terpineol	t4	6	1199		1143	S. N. L	59, 93, 121
22.62	α-Cedrene	t5	8	1421	1556	1403	N. L	119, 93, 105
22.82	β-Cedrene	t6	9	1429	1560	1403	N. L	161, 69, 204
27.34	Cedrol	t7	13	1615	2112	1543	N. L	
	Aldehydes							
10.73	Octanal	a1	1	1008		1005	S. N. L	43, 56, 84
13.88	Nonanal	a2	4	1109		1104	S. N. L	57, 41, 70
16.91	Decanal	a3	7	1212		1204	S. N. L	43, 57, 70
	Hydrocarbons							
24.80	Pentadecane	h1	12	1508	1498	1512	S. N. L	57, 43, 71, 85
29.48	Heptadecane	h2	14	1709	1699	1711	S. N. L	57, 43, 71, 86
31.77	Octadecane	h5	15	1816	1799	1852	S. N. L	57, 71, 85, 43
	Ester							
24.03	Ethyl 2(*E*)-decenoate	e1	11	1489	1758	1389	N. L	43, 55, 73
	Ketone							
23. 65	(*E*)-Geranylacetone	k1	10	1454		1420	N. L	43, 41, 69

^a^ RT: Retention time (min); ^b^ Code: antonomasia of the compound; ^c^ LRI Calc: Linear retention index calculated through n-alkanes; ^d^ LRI Nis: Linear retention index reported in the NIST Chemistry Web Book 2005; ^e^ The reliability of the identification or structural proposal is indicated by the following: (S) mass spectrum and retention time consistent with those of an authentic standard; (N) structural proposals given on the basis of mass spectral data (NIST98); (L) mass spectrum consistent with spectra found in literature. DB5: capillary column type; BPX5: apillary column type.

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
