# Peer review of "Volatiles from Different Instars of Honeybee Worker Larvae and Their Food"

_insects, 2019, doi:10.3390/insects10040118_

Round 1

Reviewer 1 Report

GC-MS Investigation of Volatiles from Honeybee Worker Larvae and Larval Food at Different Instars

            The authors describe experiments in which solid phase dynamic extraction was used to sample the headspace of honey bee larvae to investigate volatile profiles. To date, very few volatiles of larvae have been determined, and this research will lay a foundation for future studies on the activity of these compounds. I recommend this manuscript for publication with minor revisions.

            The methods seem to be well executed with appropriate controls, with just a few areas that require clarification. First, the authors use solid phase dynamic extraction, yet the reference in which these methods are discussed in detail are incorrectly labeled, i.e., there is no “Castro” in the listed references, and the in-text citation number [13] references a paper by Carson et al. in which stir bar sorptive extraction is used – quite a different procedure. Second, the authors sample from 2nd-, 4th-, and 5th-instar larvae, yet no rationale is provided on why 3rd instar larvae were not used. This must be addressed. Finally, the numbers of 2nd-, 4th-, and 5th-instar larvae used in the experiment were unequal, and leave room to doubt the interpretation of the data. Specifically, the fact that 20 2nd-instar larvae were used while only five 5th-instar larvae were used is concerning. Please address why these numbers are so disparate, or else sample more 5th-instar larvae. Five is far too low an n-value to get reliable results with an ANOVA.

 Additional critiques that should be addressed are listed below by line number.

Line 20: Revise to read “consisting of terpenoids, aldehydes, hydrocarbons, an ester, and a ketone.”

Line 26: Replace “mastered” with “determined”

Line 34-35: As written, this sentence suggests that antennae are used to disperse honey bee pheromone. Revise to read, “Volatiles in honey colonies are dispersed by bees and received through antennal reception, sometimes over long distances.”

Line 35: Replace “are” with “is an”

Line 70: I am confused by the phrase “Oviposition frames were reared.” Do the authors mean that bees were reared on oviposition frames? It is unclear.

Line 71-72: Remove the phrase for each replicate. It is redundant when considering the next sentence.

Line 72: Replace last sentence of the paragraph with “Caging events were carried out seven times, for a total of 21 combs.”

Line 93-94: The use of dashes here obfuscates the methods used. Instead, use full sentences to read, “an extraction volume of 2 mL, agitator temperature of 35 deg C, headpspace syringe temperature of 35 deg C, 30 strokes, and filling/ejecting speed of 25 uL/s.” In fact, the methods for the GC-MS analysis in the following paragraph is much easier to understand.

Line 125-126: Again, the English is a little awkward. Revise to read, “Figures were created using SigmaPlot version 12 for Windows (Systat Software Inc., California USA).”

Figure 1: The shadow effect on this figure is distracting and unnecessary. Please remove the shadow.

Table 1: The far right column should be labeled “Diagnostic ions.”

Line 144: Replace “of” with “collected from,” to read: “the largest group collected from larvae”

Line 145-148: Was there another compound? If not, the sentence should read, "while the major terpenoid of larvae was (E)-beta-ocimene (t2), accounting for 60.9-93.8%.”

Line 149-154: This section of the results is particularly unclear. I have suggested a revision below based on how I understand the authors’ results, but I may be mistaken, so please check this!
“When the amount of terpenoids was calculated as ng/individual larva as a function of larval weight, there was a continuous decrease in (E)-β-ocimene (Figure 3, column t2), α-terpineol (Figure 3, column t4), and all terpenoids together (Figure 3, column St). Two compounds disappeared completely at the 5th instar: (Z)-β-ocimene (Figure 3, column t1) and (E, Z)-alloocimene (Figure 3, cloumn t3). No significant differences were observed between instars for α-cedrene (Figure 3, colun t5) and β-cedrene (Figure 3, column t6).”

Line 177-178: replace “only changed slightly and non-significantly” with “did not change significantly.”

Figure 3: Figures with breaks on the Y-axis do not have any scale below the break. These figures should be revised to include a scale below the break in the Y-axis.

Line 202-205: This paragraph is literally repeated word for word from the previous paragraph. Please remove.

Line 235: Replace “interpreting” with “for interpretation of”

Line 244: add “instar” to read, “Samples of 2nd-instar larvae located at…”

Line 247: Replace “conservative” with “conserverd”

Line 269: Italicize the Latin name Tagetes minuta.

Line 284: Replace “including” with “and include”

Lines 293 & 307: Replace “larva” with “larvae”

Line 308-309: For clarity, please replace this sentence with, “No differences in the concentrations of this compound were found in food at different stages, either.”

Line 311: Replace “roots” with “origins”

Author Response

Response to Reviewer 1 comments

Point 1: The methods seem to be well executed with appropriate controls, with just a few areas that require clarification. First, the authors use solid phase dynamic extraction, yet the reference in which these methods are discussed in detail are incorrectly labeled, i.e., there is no “Castro” in the listed references, and the in-text citation number [13] references a paper by Carson et al. in which stir bar sorptive extraction is used – quite a different procedure.

Response 1: We have replaced Carson et al., 2009 with Castro et al., 2015” in the listed references (L.408-409).

Point 2: Second, the authors sample from 2nd-, 4th-, and 5th-instar larvae, yet no rationale is provided on why 3rd instar larvae were not used. This must be addressed.

Response 2: We essentially intended to sample from 2nd-, 3rd-, 4th- and 5th-instar larvae, but the owner of SPDE-GC-MS instrument only allowed us to use it for 3 larval development instars due to the schedule conflict . Because it was supposed to have bigger differences between 2nd- and 4th-instar larvae than between 2nd- and 3rd-instar larvae, and it is of great importance to determine the change from 4th- to 5th-instar larvae for the development of Varroa mite-control technology. So we chose 2nd-, 4th-, and 5th-instar larvae.

Point 3: Finally, the numbers of 2nd-, 4th-, and 5th-instar larvae used in the experiment were unequal, and leave room to doubt the interpretation of the data. Specifically, the fact that 20 2nd-instar larvae were used while only five 5th-instar larvae were used is concerning. Please address why these numbers are so disparate, or else sample more 5th-instar larvae. Five is far too low an n-value to get reliable results with an ANOVA.

Response 3: We actually used 10 5th-instar larvae, but we mistyped the number with “5”. We have corrected it in the text (L. 83).

Point 4: Line 20: Revise to read “consisting of terpenoids, aldehydes, hydrocarbons, an ester, and a ketone.”

Response 4: We have revised the sentence (L. 20-21).

Point 5: Line 26: Replace “mastered” with “determined”

Response 5: We have replaced “mastered” with “determined” (L. 27).

Point 6: Line 34-35: As written, this sentence suggests that antennae are used to disperse honey bee pheromone. Revise to read, “Volatiles in honey colonies are dispersed by bees and received through antennal reception, sometimes over long distances.”

Response 6: We have rewritten the sentence (L. 36 Volatiles in honey bee colonies are easily received by bees through antennal reception, sometimes over long distances.).

Point 7: Line 35: Replace “are” with “is an”

Response 7: We have replace “are” with “is an” (L. 36).

Point 8: Line 70: I am confused by the phrase “Oviposition frames were reared.” Do the authors mean that bees were reared on oviposition frames? It is unclear.

Response 8: We have reworded the sentence to make it clear (L. 73-74).

Point 9: Line 71-72: Remove the phrase for each replicate. It is redundant when considering the next sentence.

Response 9: We have removed the phrase “for each replicate” (L. 75).

Point 10: Line 72: Replace last sentence of the paragraph with “Caging events were carried out seven times, for a total of 21 combs.”

Response 10: We have replaced the last sentence of this paragraph (L. 75-76).

Point 11: Line 93-94: The use of dashes here obfuscates the methods used. Instead, use full sentences to read, “an extraction volume of 2 mL, agitator temperature of 35 deg C, headpspace syringe temperature of 35 deg C, 30 strokes, and filling/ejecting speed of 25 uL/s.” In fact, the methods for the GC-MS analysis in the following paragraph is much easier to understand.

Response 11: We have used full sentences instead of dashes (L. 97-99).

Point 12: Line 125-126: Again, the English is a little awkward. Revise to read, “Figures were created using SigmaPlot version 12 for Windows (Systat Software Inc., California USA).”

Response 12: We have reworded the sentence as the reviewer’s comment (L. 129-131).

Point 13: Figure 1: The shadow effect on this figure is distracting and unnecessary. Please remove the shadow.

Response 13: We have removed the shadow effect from Figure 1 (L. 136).

Point 14: Table 1: The far right column should be labeled “Diagnostic ions.”

Response 14: We have replaced “Quantifier” with “Diagnostic” in Table 1.

Point 15: Line 144: Replace “of” with “collected from,” to read: “the largest group collected from larvae”

Response 15: We have replaced “of” with “collected from” (L. 149).

Point 16: Line 145-148: Was there another compound? If not, the sentence should read, "while the major terpenoid of larvae was (E)-beta-ocimene (t2), accounting for 60.9-93.8%.”

Response 16: We have replaced “terpenoids” with “terpenoid”, and deleted the word “and” from the sentence (L. 153).

Point 17: Line 149-154: This section of the results is particularly unclear. I have suggested a revision below based on how I understand the authors’ results, but I may be mistaken, so please check this!

When the amount of terpenoids was calculated as ng/individual larva as a function of larval weight, there was a continuous decrease in (E)-β-ocimene (Figure 3, column t2), α-terpineol (Figure 3, column t4), and all terpenoids together (Figure 3, column St). Two compounds disappeared completely at the 5th instar: (Z)-β-ocimene (Figure 3, column t1) and (E, Z)-alloocimene (Figure 3, cloumn t3). No significant differences were observed between instars for α-cedrene (Figure 3, colun t5) and β-cedrene (Figure 3, column t6).”

Response 17: We have rewritten this section of the results according to the reviewer’s advises (L. 155-164).

Point 18: Line 177-178: replace “only changed slightly and non-significantly” with “did not change significantly.”

Response 18: We have replaced “only changed slightly and non-significantly” with “did not change significantly” (L. 189).

Point 19: Figure 3: Figures with breaks on the Y-axis do not have any scale below the break. These figures should be revised to include a scale below the break in the Y-axis.

Response 19:  A scale has been included below the break in the Y-axis in Figure 3.

Point 20: Line 202-205: This paragraph is literally repeated word for word from the previous paragraph. Please remove.

Response 20: We have removed the repeated paragraph (L. 196-200).

Point 21: Line 235: Replace “interpreting” with “for interpretation of”

Response 21: We have replace “interpreting” with “for interpretation of” (L. 248).

Point 22: Line 244: add “instar” to read, “Samples of 2nd-instar larvae located at…”

Response 22: We have added the word “instar” to the sentence “Samples of 2nd-instar larvae located at…” (L. 258).

Point 23: Line 247: Replace “conservative” with “conserverd”

Response 23: We have replace “conservative” with “conserved” (L. 261).

Point 24: Line 269: Italicize the Latin name Tagetes minuta.

Response 24: We have italicized the Latin name Tagetes minuta (L. 289).

Point 25: Line 284: Replace “including” with “and include”

Response 25: We have replaced “including” with “They include” (L. 308-309).

Point 26: Lines 293 & 307: Replace “larva” with “larvae”

Response 26: We have replaced “larva” with “larvae” (L. 319 & 343).

Point 27: Line 308-309: For clarity, please replace this sentence with, “No differences in the concentrations of this compound were found in food at different stages, either.”

Response 27: We have replaced this sentence with, “No differences in the concentrations of this compound were found in food at different stages, either.” (L. 344-345).

Point 28:  Line 311: Replace “roots” with “origins”

Response 28: We have replaced “roots” with “origins” (L. 347).

Reviewer 2 Report

This article deals with the discovery of new volatile compounds coming from developing honeybees. This field is of course widely important in not only understanding the basic biology of bees, but for developing future interventions for the benefit of bees and agricultural economy. The article is well written and takes an interesting and experimentally appropriate approach for showing the changes in volatile profiles with respect to ontogeny. However, there are several problems and concerns that require major revision before the article can be fully considered for publication.

In no particular order:

-The writing is quite good for the most part. There are a few spots where the word choice and grammar are wrong. I suggest that the authors construct a few more drafts to alleviate those problems, or have a native English speaker edit the manuscript. Here are a few that jumped out at me:

Line 26 “measured” not “mastered”?

Line 26 “volatile” not “volatiles’”

Line 65 “strains” not “strain”

Line 80/81 consider rewriting this sentence. “To monitor changes in volatiles with respect to ontogeny”

Line 67 Should be “39°” not “39o”

Line 321 “Set of rules” sounds strange…Perhaps “follows a natural progression”?

-To boil this manuscript down to its core, the authors have essentially taken a more sensitive GC-MS system to discovers some new volatiles associated with developing bees. Ergo, Line 61-63 is an overstatement, this article is NOT the first steps towards understanding the function of volatiles. With all due respect, I think the authors must soften the tone in this case.

-Unless I honestly missed it is there a reason why 3rdinstar larvae were skipped in the experimental design? I don’t think it’s a disqualifying problem…However, the authors do not delve into why it was skipped.

-The discussion is deficiently short, and there is little information that gives any insight on what the possible roles of these volatiles are in development. The authors only delve slightly into the role of these different classes of compounds in honeybees. What about across all holometabolous insects? Are there any corollaries in other social or asocial insect systems?

-Ethyl 2(E)-decenoate might be a product of incomplete beta oxidation of longer fatty acids? Why would this volatile intermediate accumulate if it is merely a transient metabolite? This explanation doesn’t make sense. Are honeybees even undergoing large scale beta-oxidation during the larval stage?

Line 299 Tentatively identified? Did the GC-MS predict multiple possible molecules from that peak? I’m confused, could this compound be something different?

Line 306 This statement is false, and is another indicator that the authors have not properly mined the literature to support their claims. See: https://link.springer.com/article/10.1007/s13592-015-0358-xMattila’s group studied geranyl acetones during queen flight versus in-hive.

Line 311-317 is good, but how do your localities compare to other published volatile studies? This should be discussed.

In closing the article does merit publication, but the authors need to tighten up the writing a little and expound more in the discussion.

Author Response

Response to Reviewer 2 Comments

Point 1: -The writing is quite good for the most part. There are a few spots where the word choice and grammar are wrong. I suggest that the authors construct a few more drafts to alleviate those problems, or have a native English speaker edit the manuscript. Here are a few that jumped out at me:

Response 1: We have made the manuscript edited by the highly qualified native

English speaking editors at American Journal Experts.

Point 2: Line 26 “measured” not “mastered”?

Response 2: We have replaced “mastered” with “determined” (L. 27).

Point 3: Line 26 “volatile” not “volatiles’”

Response 3: We have rewritten this sentence (L. 27).

Point 4: Line 65 “strains” not “strain”

Response 4: We have replaced “strain” with “strains” (L. 68).

Point 5: Line 80/81 consider rewriting this sentence. “To monitor changes in volatiles with respect to ontogeny”

Response 5: We have rewritten this sentence according to the reviewer’s advice (L. 85).

Point 6: Line 67 Should be “39°” not “39o”

Response 6: We have replaced “39o” with “39°” (L. 70).

Point 7: Line 321 “Set of rules” sounds strange…Perhaps “follows a natural progression”?

Response 7: We have replaced “have a set of rules” with “follow certain regularity” (L. 361).

Point 8: -To boil this manuscript down to its core, the authors have essentially taken a more sensitive GC-MS system to discovers some new volatiles associated with developing bees. Ergo, Line 61-63 is an overstatement, this article is NOT the first steps towards understanding the function of volatiles. With all due respect, I think the authors must soften the tone in this case.

Response 8: We have deleted the last sentence of this paragraph (L. 65-66).

Point 9:-Unless I honestly missed it is there a reason why 3rdinstar larvae were skipped in the experimental design? I don’t think it’s a disqualifying problem…However, the authors do not delve into why it was skipped.

Response 9: We essentially intended to sample from 2nd-, 3rd-, 4th- and 5th-instar larvae larvae, but the owner of SPDE-GC-MS instrument only allowed us to use it for 3 larval development instars, due to the schedule conflict . Because it was supposed to have bigger differences between 2nd- and 4th-instar larvae than between 2nd- and 3rd-instar larvae; and it is of great importance to determine the change from 4th- to 5th-instar larvae for the development of Varroa mite-control technology. So we chose 2nd-, 4th-, and 5th-instar larvae.

Point 10: -The discussion is deficiently short, and there is little information that gives any insight on what the possible roles of these volatiles are in development. The authors only delve slightly into the role of these different classes of compounds in honeybees. What about across all holometabolous insects? Are there any corollaries in other social or asocial insect systems?

Response 10: We have added some information to discuss the role of these different classes of compounds in other insects (L. 271-346).

Point 11: -Ethyl 2(E)-decenoate might be a product of incomplete beta oxidation of longer fatty acids? Why would this volatile intermediate accumulate if it is merely a transient metabolite? This explanation doesn’t make sense. Are honeybees even undergoing large scale beta-oxidation during the larval stage?

Response 11: We have changed the statement (L. 334-338).

Point 12: Line 299 Tentatively identified? Did the GC-MS predict multiple possible molecules from that peak? I’m confused, could this compound be something different?

Response 12: One chromatographic peak could predict several possible compounds (especially isomers), based on the data from the library of the GC-MS software. For confirming the identities of the compound(s) we should compare the measured Rt (retention time) and MS (mass spectra) of the unknown compound(s), with Rt and MS of the authentic standard(s).

Point 13: Line 306 This statement is false, and is another indicator that the authors have not properly mined the literature to support their claims. See: https://link.springer.com/article/10.1007/s13592-015-0358-xMattila’s group studied geranyl acetones during queen flight versus in-hive.

Response 13: We have corrected this false statement (L. 339).

Point 14: Line 311-317 is good, but how do your localities compare to other published volatile studies? This should be discussed.

Response 14: In this paragraph, we discussed the weakness of the present study and the prospects of the future research. We would like to compare different honeybee worker larvae samples from different localities based on the volatile study. This reviewer offered another way of thinking to compare the volatiles of worker larvae with the volatiles published by other studies based on the same locality. We have added this information in the last paragraph of discussion (L. 347-355).

Round 2

Reviewer 2 Report

Yes, the discussion now captures the role of these different volatile functional groups not only in honey bees, but across Insects. Additionally, the English and grammar far exceeds even my abilities.